# Transient Blockade of Type I Interferon Signalling Promotes Replication of Dengue Virus Strain D2Y98P in Adult Wild-Type Mice

**DOI:** 10.3390/v15040814

**Published:** 2023-03-23

**Authors:** Lucas Wilken, Sonja Stelz, Chittappen Kandiyil Prajeeth, Guus F. Rimmelzwaan

**Affiliations:** Research Center for Emerging Infections and Zoonoses (RIZ), University of Veterinary Medicine (TiHo), 30559 Hannover, Germany

**Keywords:** dengue virus, mouse model, host restriction, vaccine, antiviral

## Abstract

Dengue virus serotypes 1 to 4 (DENV1–4) place nearly half the global population at risk of infection and the licenced tetravalent dengue vaccine fails to protect individuals who have not previously been exposed to DENV. The development of intervention strategies had long been hampered by the lack of a suitable small animal model. DENV does not replicate in wild-type mice due to its inability to antagonise the mouse type I interferon (IFN) response. Mice deficient in type I IFN signalling (*Ifnar1*^−/−^ mice) are highly susceptible to DENV infection, but their immunocompromised status makes it difficult to interpret immune responses elicited by experimental vaccines. To develop an alternative mouse model for vaccine testing, we treated adult wild-type mice with MAR1-5A3—an IFNAR1-blocking, non-cell-depleting antibody—prior to infection with the DENV2 strain D2Y98P. This approach would allow for vaccination of immunocompetent mice and subsequent inhibition of type I IFN signalling prior to challenge infection. While *Ifnar1*^−/−^ mice quickly succumbed to infection, MAR1-5A3-treated mice did not show any signs of illness but eventually seroconverted. Infectious virus was recovered from the sera and visceral organs of *Ifnar1*^−/−^ mice, but not from those of mice treated with MAR1-5A3. However, high levels of viral RNA were detected in the samples of MAR1-5A3-treated mice, indicating productive viral replication and dissemination. This transiently immunocompromised mouse model of DENV2 infection will aid the pre-clinical assessment of next-generation vaccines as well as novel antiviral treatments.

## 1. Introduction

Dengue virus (DENV), a mosquito-borne flavivirus related to Zika virus (ZIKV), West Nile virus (WNV), and yellow fever virus (YFV), exists as four distinct serotypes (DENV1–4) that cause an estimated 390 million infections each year, including 96 million symptomatic cases [1]. The clinical spectrum of symptomatic DENV infection ranges from a mild febrile illness, termed dengue fever (DF), to severe disease forms, namely dengue haemorrhagic fever (DHF) and dengue shock syndrome (DSS), the hallmarks of which are thrombocytopenia, coagulopathy, and vascular leakage [2]. Individuals experiencing secondary infection with a different (heterologous) serotype develop severe dengue disease more frequently [3,4]. This is thought to be the result of antibody-dependent enhancement (ADE), a phenomenon in which cross-reactive antibodies induced during primary infection bind but fail to neutralise the infecting heterologous serotype and mediate its uptake into Fc receptor–expressing myeloid cells [5,6,7,8,9]. ADE also likely underlies the poor long-term safety of the licenced dengue vaccine CYD-TDV (Dengvaxia) in immunologically naïve populations, which were found to be at higher risk for hospitalisation upon subsequent DENV exposure [10,11,12]. Alternative strategies are therefore being investigated for the generation of a safe and effective dengue vaccine.

Over the past few decades, extensive efforts have been made to develop mouse models of DENV infection for studying viral pathogenesis and evaluating intervention strategies [13]. Early on, it was found that DENV replicates poorly and does not cause clinical signs of disease in wild-type mice [14], which later was shown to be partly due to its failure to antagonise type I interferon (IFN) induction through STING (stimulator of interferon genes) and type I IFN signalling through STAT2 (signal transducer and activator of transcription 2) in mouse cells [15,16]. Most mouse models of DENV infection therefore use mice lacking interferon-alpha/beta receptor subunit 1 (IFNAR1; *Ifnar1^−/−^* mice) or double-knockout mice that are additionally deficient in type II IFN signalling due to the absence of interferon-gamma receptor subunit 1 (IFNGR1; *Ifnar1^−/−^ Ifngr1^−/−^* mice) [17,18,19,20,21,22]. While these models recapitulate key features of dengue disease seen in humans, such as vascular leakage, liver injury, thrombocytopaenia, and cytokine storm, their immunocompromised status impairs the translatability of immune responses induced by experimental vaccines to immunocompetent hosts. This is because type I IFN not only promotes initial viral clearance through the induction of interferon-stimulated genes (ISGs), but also enhances other important aspects of antiviral immunity, including B-cell responses [23,24,25] and isotype class switching [25,26], cross-presentation [27,28,29], CD4^+^ T-cell activation and expansion [30,31], as well as CD8^+^ T-cell maturation, expansion, and memory formation [32,33,34]. More recently, mice with conditional deletions of *Ifnar1* in different subsets of myeloid cells (i.e., *LysM-Cre*^+/−^
*Ifnar1^fl/fl^* and *CD11c-Cre*^+/−^
*Ifnar1^fl/fl^* mice) have been established as more immunocompetent models of DENV infection [35,36]. However, these mice are not widely available and thus do not allow for high-throughput testing of dengue vaccine candidates. Alternatively, type I IFN signalling can be transiently suppressed in wild-type mice using the IFNAR1-blocking, non-cell-depleting monoclonal antibody MAR1-5A3 [37], which has been shown to render them susceptible to infection with low-passage clinical isolates of WNV [34], ZIKV [38,39,40], and DENV1 [41]. This approach allows for vaccination in immunocompetent animals with subsequent attenuation of type I IFN signalling at the time of viral challenge.

In this study, we investigated whether MAR1-5A3 pre-treatment could promote replication and pathogenesis of the non-mouse-adapted DENV2 strain D2Y98P in adult wild-type mice. We found that transient blockade of type I IFN signalling in these animals did not reproduce the disease phenotype observed in *Ifnar1^−/−^* mice. While no infectious virus was detected in the sera and visceral organs of MAR1-5A3-treated mice, high levels of viral RNA were present in these samples, indicating productive viral replication and dissemination. We thus provide a tractable mouse model for high-throughput evaluation of next-generation dengue vaccines and other antiviral intervention strategies.

## 2. Materials and Methods

### 2.1. Ethics Statement

All animal experiments were performed in strict compliance with European guidelines (EU directive on animal testing 2010/63/EU) and the German Animal Welfare Act. The experimental protocols were approved by Lower Saxony State Office for Consumer Protection and Food Safety (LAVES, Oldenburg, Germany—33.9-42502-04-21/3806).

### 2.2. Mice

Specific-pathogen-free (SPF) female C57BL/6J mice were purchased from Charles River Laboratories (Goettingen, Germany) and congenic female *Ifnar1*^−/−^ mice were generously provided by Markus Keller (Friedrich Loeffler Institute, Riems, Germany). All mice were 10 to 12 weeks old at the start of the experiments. The animals were housed under SPF conditions in individually ventilated cage (IVC) systems at the Research Center for Emerging Infections and Zoonoses of the University of Veterinary Medicine, Hannover, Germany, and were given food and water ad libitum.

### 2.3. Cells and Virus

Vero (ATCC CCL-81) and BHK-21 (ATCC CCL-10) cells were cultured in Eagle’s minimum essential medium (EMEM; Sigma-Aldrich, St. Louis, MO, USA) supplemented with 10% heat-inactivated foetal bovine serum (FBS; Gibco, Waltham, MA, USA), 1% penicillin/streptomycin (P/S; Sigma-Aldrich, St. Louis, MO, USA), 20 mM HEPES (Gibco, Waltham, MA, USA), and 1% GlutaMAX (Gibco, Waltham, MA, USA), and kept at 37 °C and 5% CO_2_. C6/36 cells were maintained in Leibovitz’s L-15 medium (Gibco, Waltham, MA, USA) supplemented with 10% FBS, 10% tryptose phosphate broth (Gibco, Waltham, MA, USA), 1% P/S, 20 mM HEPES, and 1% GlutaMAX, and kept at 28 °C in the absence of CO_2_.

DENV2 strain D2Y98P (hereafter termed D2Y98P; kindly provided by Sylvie Alonso, National University of Singapore, Singapore), originally isolated from a dengue patient in Singapore in 1998 and passaged about 20 times in C6/36 cells [18], was propagated in C6/36 cells. Virus was concentrated and buffer-exchanged into PBS (pH 7.4) using Amicon Ultra-15 centrifugal filter units with 100,000 Dalton molecular weight cut-off (Millipore, Burlington, MA, USA) and titrated by plaque assay in BHK-21 cells as well as by focus-forming assay in Vero cells.

### 2.4. Mouse Infections

Mice were infected subcutaneously with 10^6^ plaque-forming units (PFU) of D2Y98P. Wild-type mice received a single intraperitoneal dose of 2 mg of either IFNAR1-blocking antibody (mouse anti-mouse IFNAR1, IgG1, clone MAR1-5A3, Leinco Technologies, Fenton, MO, USA) [37] or isotype control antibody (mouse anti-human IFNGR1, IgG1, clone GIR-208, Leinco Technologies, Fenton, MO, USA) [37] one day prior to infection with D2Y98P. Mice were assessed daily for changes in weight loss (score of 0, none; 1, ≥5%; 2, ≥15%; 3, ≥20%), appearance (0, normal; 1, ruffled fur; 2, hunching; 3, severe hunching and closed eyes), and activity (0, normal; 1, moderately reduced activity; 2, lethargy; 3, severe lethargy), as well as the presence of gastrointestinal symptoms (0, none; 1, diarrhoea or bloating; 2, diarrhoea and bloating). Mice were euthanised when reaching a total score of 7, or a score of 3 in one of the first three categories (weight loss, appearance, or activity).

To avoid any distress caused by daily blood collection, only alternate mice were bled at each time point. Blood was collected into MiniCollect Z Serum Sep tubes (Greiner Bio-One, Kremsmünster, Austria) and serum was separated by centrifugation at 3000× *g* for 10 min at RT and subsequently stored at −80 °C. Organs were collected in 1 mL of infection medium (EMEM supplemented with 2% FBS, 1% P/S, 20 mM HEPES and 1% GlutaMAX) and homogenised for 1 min at 30 Hz using a TissueLyser II device (QIAGEN, Hilden, Germany). Homogenates were clarified by centrifugation at 17,000× *g* for 10 min at 4 °C and subsequently stored at −80 °C.

### 2.5. Virus Titrations

Infectious virus in serum was quantified by focus-forming assay in Vero cells. Serum samples were tenfold serially diluted in infection medium, then 50 µL of the diluted samples was applied in duplicate to Vero cells (seeded the day before in flat-bottom 96-well plates at 10^4^ cells per well) and incubated for 1 h at 37 °C. Afterwards, the inoculum was removed and overlay medium (1% carboxymethyl cellulose [CMC; Sigma-Aldrich, St. Louis, MO, USA] in Temin’s modified Eagle medium [MEM; Gibco, Waltham, MA, USA], supplemented with 2% FBS, 1% P/S, 20 mM HEPES, and 1% GlutaMAX) was added, and the cells were incubated for 3 days at 37 °C. Upon completion of the incubation period, the overlay medium was aspirated, and the cells were fixed with 4% paraformaldehyde (PFA) in PBS for 15 min at RT, permeabilised using 0.5% Triton X-100 in PBS for 10 min at RT and blocked using 2.5% normal horse serum (Cytiva, Marlborough, MA, USA) in PBS for 30 min at RT. The cells were stained with chimeric rabbit monoclonal anti-flavivirus group antigen antibody (1:2000; clone D1-4G2-4-15, antibodies-online) for 1 h at RT, followed by Alexa Fluor 488-conjugated donkey anti-mouse IgG antibody (1:500; Invitrogen, Waltham, MA, USA) for 1 h at RT in the dark. Both antibodies were diluted in 2.5% normal horse serum in PBS. Cells were washed three times with PBS after each antibody incubation. Foci were counted manually using a Leica DMi8 microscope (Leica Microsystems, Wetzler, Germany).

Infectious virus in organs was quantified by plaque assay in BHK-21 cells. Briefly, BHK-21 cells were seeded the day before in flat-bottom 24-well plates (Corning) at a density of 1.25 × 10^5^ cells per well. Organ homogenates were tenfold serially diluted in infection medium, then 200 µL of the diluted samples was applied in duplicate to the cells and incubated for 1 h at 37 °C. Afterwards, the inoculum was removed, overlay medium was added, and the cells were incubated for 4 days at 37 °C. Upon completion of the incubation period, the overlay medium was aspirated, the cells were fixed with 4% paraformaldehyde (PFA) in PBS for 15 min at RT and subsequently stained with 1% crystal violet (Fisher Scientific, Waltham, MA, USA) for 10 min at RT. Plates were thoroughly rinsed, then dried, and plaques were scored visually.

### 2.6. Neutralisation Assay

Serum samples were thawed, heat-inactivated for 60 min at 56 °C, and then twofold serially diluted in serum-free EMEM from a starting dilution of 1:10. Diluted serum samples were mixed with an equal volume of 100 focus-forming units (FFU)/well of D2Y98P and incubated for 1 h at 37 °C. Subsequently, 50 µL of the mixtures was applied in duplicate to Vero cells (seeded the day before in flat-bottom 96-well plates at 10^4^ cells/well) and incubated for 1 h at 37 °C. Afterwards, the inoculum was removed, overlay medium was added, and the cells were incubated for 3 days at 37 °C. Foci were visualised and enumerated as described above. Fifty percent focus reduction neutralisation test (FRNT_50_) values were determined by non-linear regression analysis using Prism software version 9.5.0 (GraphPad, San Diego, CA, USA).

### 2.7. Viral RNA Quantification

Viral RNA was isolated from 10 µL of serum (adjusted to 140 µL with PBS) or 140 µL of organ homogenates using the QIAamp Viral RNA Mini Kit (QIAGEN, Hilden, Germany), eluted in 60 µL AVE buffer and stored at −80 °C until use. Viral RNA copies were determined using the Luna One-Step Universal Probe RT-qPCR Kit (New England Biolabs, Ipwich, MA, USA), 5 µL of RNA, 600 nM of forward primer (DENV2-8597-FW, 5′-TGACAGACACGACTCCATTTG-3′), 600 nM of reverse primer (DENV2-8710-RV, 5′-GCCAYTCTGCCGTGATTT-3′), and 300 nM of probe (DENV2-probe, 5′-FAM-AACCCA AGAACCGAAAGAAGGCAC-BHQ1-3′). Oligonucleotides were synthesised by Microsynth (Balgach, Switzerland). Reactions were performed on a CFX96 Touch real-time PCR detection system (Bio-Rad, Hercules, CA, USA) using the following thermal profile: 15 min at 55 °C, 3 min at 95 °C, followed by 45 cycles of 15 s at 95 °C, 1 min at 60 °C, and plate read. Data were analysed using CFX Maestro software (Bio-Rad, Hercules, CA, USA) and copy numbers were calculated using a standard curve generated from a gel-purified PCR product of the target region.

### 2.8. Statistical Analysis

Groups were compared using Mann–Whitney *U* test or unpaired t-test with Welch’s correction, as specified in the figure legends. *p* values < 0.05 were considered statistically significant. All statistical analyses were performed using Prism software version 9.5.0 (GraphPad, San Diego, CA, USA).

## 3. Results

### 3.1. D2Y98P Infection Is Asymptomatic in Adult Wild-Type Mice Treated with IFNAR1-Blocking Antibody

We first tested wild-type C57BL/6J mice and congenic *Ifnar1*^−/−^ mice for their susceptibility to disease caused by DENV2 strain D2Y98P, a non-mouse-adapted virus derived from a Singaporean clinical isolate [18]. The animals were 10 to 12 weeks old at the start of the experiment, which corresponds to their age at the time of viral challenge following a typical prime–boost vaccination regimen. One day prior to infection, C57BL/6J mice were injected intraperitoneally with a saturating dose (i.e., 2 mg) of the IFNAR1-blocking monoclonal antibody MAR1-5A3 [37]. Another group of C57BL7/6J mice received 2 mg of an isotype control antibody (GIR-208), which targets human interferon-gamma receptor subunit 1 (IFNGR1) [42]. Mice were inoculated subcutaneously with 10^6^ plaque-forming units (PFU) of D2Y98P and monitored daily for 15 days for weight loss, clinical symptom development, and survival.

As expected, *Ifnar1*^−/−^ mice showed rapid weight loss, developed clinical signs of disease (including one animal with diarrhoea-like symptoms), and were euthanised at 4 or 5 dpi because they reached a pre-defined humane endpoint (Figure 1a–c). At necropsy, these animals presented with splenomegaly and liver lesions. By contrast, C57BL/6J mice treated with MAR1-5A3 or GIR-208 did not lose weight, displayed no signs of illness, and survived until the end of the experiment (i.e., 15 dpi). To assess whether these animals had been asymptomatically infected, we tested for seroconversion to D2Y98P by measuring virus-neutralising antibody titres in sera collected at 15 dpi. No virus neutralisation was observed for the sera of GIR-208-treated mice, whereas the sera of MAR1-5A3-treated mice exhibited substantial neutralising activity (Figure 1d). Together, these findings indicate that antibody-mediated blockade of type I IFN signalling enables amplification of D2Y98P in wild-type mice but does not recapitulate the disease phenotype seen in *Ifnar1^−/−^* mice.

### 3.2. Infectious Virus Is Undetectable in Sera and Organs of D2Y98P-Infected Adult Wild-Type Mice Treated with IFNAR1-Blocking Antibody

After demonstrating asymptomatic D2Y98P infection in MAR1-5A3-treated mice, we next determined the kinetics of viraemia and viral dissemination. Adult mice were treated as before and inoculated subcutaneously with 10^6^ PFU of D2Y98P. Sera were collected at 2, 3, and 5 dpi, organs (spleen, kidney, jejunum, and liver) were harvested at 5 dpi, and tested for the presence of infectious virus. Two out of four *Ifnar1*^−/−^ mice had to be euthanised at 4 dpi because humane endpoint criteria were met. As before, gross pathological examination at necropsy demonstrated splenomegaly and liver lesions in all *Ifnar1*^−/−^ mice. Interestingly, we also observed splenomegaly but no other gross pathological changes in mice treated with MAR1-5A3, whereas the organs of GIR-208-treated mice appeared normal. *Ifnar1*^−/−^ mice had comparable levels of viraemia at 2 and 3 dpi, but virus was seemingly cleared from the circulation by 5 dpi (Figure 2a). No infectious virus was detected in the sera of mice treated with MAR1-5A3 or GIR-208 at any time point. Spleens and kidneys of the *Ifnar1*^−/−^ mice contained similar viral titres (Figure 2b,c), while virus was only detectable in the jejunum of one of these animals (Figure 2d). For one out of six mice treated with MAR1-5A3, virus was found in the spleen, though this was barely above the limit of detection. No infectious virus was detected in the kidneys or jejunum of these animals. All organs harvested from GIR-208-treated mice tested negative for virus. Unfortunately, virus titrations could not be performed for liver homogenates as they were highly cytotoxic to BHK-21 cells. These results suggest that antibody-mediated blockade of type I IFN signalling is insufficient for the production of detectable levels of infectious D2Y98P in wild-type mice.

### 3.3. High Levels of Viral RNA Are Present in Sera and Organs of D2Y98P-Infected Adult Wild-Type Mice Treated with IFNAR1-Blocking Antibody

As we did not find infectious virus in any of the samples taken from MAR1-5A3-treated mice, we tested whether we could at least detect viral RNA in them. Thus, the same serum and organ samples were subjected to DENV2-specific RT-qPCR. High levels of viral RNA were detected in the sera of *Ifnar1*^−/−^ mice at 2 and 3 dpi, whereas much lower levels were present at 5 dpi (Figure 3a), in line with the drop in infectious virus titres at this time point. Notably, sera collected at 2 and 3 dpi from mice treated with MAR1-5A3 also contained substantial amounts of viral RNA that were about 1000-fold lower than those observed for *Ifnar1*^−/−^ mice. The sera of most animals in this group remained positive for viral RNA at 5 dpi, though only slightly above the limit of detection. Viral RNA was undetectable in the sera of GIR-208-treated mice at all time points. *Ifnar1*^−/−^ mice had high levels of viral RNA in the spleen, kidneys, and jejunum (Figure 3a–c), and somewhat lower levels in the liver (Figure 3d). Mice treated with MAR1-5A3 displayed viral RNA in all of these organs, but its levels were about 1.5 logs lower (spleen) or 3 logs lower (kidney, jejunum, and liver) than in *Ifnar1*^−/−^ mice. All organs harvested from GIR-208-treated mice tested negative for viral RNA. Collectively, these findings suggest that antibody-mediated blockade of type I IFN signalling enabled replication of D2Y98P at the primary sites of infection and systemic viral dissemination.

## 4. Discussion

The inability of DENV to antagonise type I IFN signalling in mice [15] and its central role in priming innate and adaptive immunity [43] are two seemingly incompatible factors that impede the development of an immunocompetent dengue mouse model. The importance of type I IFN signalling in mounting a functional immune response to DENV infection and experimental vaccination was previously highlighted using conditional IFNAR1 knockout mice [35]. That study showed that *Ifnar1*^−/−^ mice were severely impaired in generating de novo CD8^+^ T-cell responses upon DENV infection, whereas mice lacking IFNAR1 only in a subset of myeloid cells (i.e., *LysM-Cre*^+/−^
*Ifnar1^fl/fl^* and *CD11c-Cre*^+/−^
*Ifnar1^fl/fl^* mice) showed a rapid increase in virus-specific CD8^+^ T cells [35]. Furthermore, despite developing comparable neutralising antibody titres after recombinant protein vaccination, *Ifnar1*^−/−^ mice were not protected against subsequent DENV challenge, which was thus thought to be due to inefficient induction of T-cell responses [35,44]. Here, we developed a transiently immunocompromised mouse model of DENV2 infection that is well suited for vaccine testing, as it allows for vaccine-induced immune responses to be elicited prior to the inhibition of type I IFN signalling by the IFNAR1-blocking antibody MAR1-5A3.

We found that MAR1-5A3 pre-treatment promoted replication and dissemination of D2Y98P, but not disease susceptibility, in wild-type mice. *Ifnar1*^−/−^ mice in both C57BL/6 and 129/Sv backgrounds develop cytokine storm, vascular leakage, and liver damage following D2Y98P infection and become moribund within a few days [35,45]. The cause of death in these animals remains elusive but is likely a combination of the aforementioned manifestations. In line with these studies, we also observed complete lethality in *Ifnar1*^−/−^ mice, which was associated with macroscopic liver lesions. By contrast, wild-type mice treated with MAR1-5A3 showed no weight loss and survived D2Y98P infection. Our second set of experiments then revealed the absence of gross pathological changes—with the exception of splenomegaly—in these mice, possibly explaining the lack of clinical signs. We believe that this might be directly related to the level of viral replication. While we did not detect infectious virus in the sera and visceral organs of MAR1-5A3-treated mice, high levels of viral RNA were present in these samples, which may stem from defective interfering particles [46]. However, viral RNA copy numbers were much higher in the sera and most tissues, including the liver, of *Ifnar1*^−/−^ mice, suggesting a direct correlation between viral burden and disease pathology. Along this line, the difference in viral RNA levels was much smaller between the spleens of *Ifnar1*^−/−^ and MAR1-5A3-treated mice, which might explain the development of splenomegaly in these animals. However, as we did not test for the induction of pro-inflammatory cytokines or the presence of inflammatory cells in infected tissues, we cannot exclude the contribution of immune-mediated disease mechanisms.

Our findings add to previous studies using transient IFNAR1 blockade to develop mouse models of flavivirus infections. This strategy was first applied to WNV, which replicates and induces lethality in wild-type mice in an age- and dose-dependent manner, and was shown to enhance WNV replication and to shorten the mean time to death [34]. Following the re-emergence of ZIKV, attempts were made to develop a mouse model for the Asian lineage ZIKV strain H/PF/2013, which replicated to low levels in a fraction of wild-type mice. MAR1-5A3 pre-treatment strongly enhanced viraemia, but did not lead to weight loss, neurologic disease, or lethality [38]. By contrast, the African lineage ZIKV strain Dakar 41525, which also produced low-level viraemia in wild-type mice, replicated to high titres in the brains of MAR1-5A3-treated mice and caused lethal disease in nearly half of them [39,40]. Complete lethality was achieved when the virus was administered intraperitoneally instead of subcutaneously [39]. Moreover, two isolates of Usutu virus (USUV), another neurotropic flavivirus, were shown to replicate efficiently and cause death in *Ifnar1*^−/−^ mice, whereas no replication of either virus was observed in outbred CD-1 mice [47]. Intriguingly, pre-treatment with MAR1-5A3 promoted lethality of one isolate but not the other, which was associated with large differences in viral load [47]. MAR1-5A3 reportedly does not cross the blood–brain barrier efficiently [34], thus the neurovirulence of these virus strains is thought to result from an enhanced capacity for invasion of and replication in the central nervous system. Another study found that the mouse-adapted DENV2 strain New Guinea C (NGC) replicated to high levels but did not cause disease in wild-type mice treated with MAR1-5A3 [48]. Notably, virus production was shown to be greater and more sustained in younger (4 weeks old) than in older (10 weeks old) animals [48]. Collectively, these and our studies indicate that multiple factors, including strain-specific virulence determinants, passage history, viral dose, inoculation route, and mouse age, influence the ability of flaviviruses to replicate and cause disease in MAR1-5A3-treated wild-type mice.

Because we sought to develop a dengue mouse model that resembles natural infection as closely as possible and that could be employed in pre-clinical vaccine testing, we chose to use a non-mouse-adapted virus strain, a peripheral route of virus inoculation, and mice of adult age. Thus, there is little room for improvement of this model without significantly deviating from our initial goal. We believe that MAR1-5A3 administration might be the only modifiable factor. In our experiments, we intraperitoneally administered a saturating dose (i.e., 2 mg) of MAR1-5A3, which has a reported half-life of 5.2 days in wild-type mice [37]. The age of the mice used in that study is unknown, but it is conceivable that higher antibody doses are required to saturate IFNAR1 in adult versus young mice. Thus, an increased dose of MAR1-5A3 might enhance viral replication and potentially promote disease development. By contrast, increasing antibody half-life through repeated administrations of MAR1-5A3 after infection is unlikely to prolong viraemia beyond 3 dpi. This is because *Ifnar1*^−/−^ mice had also largely cleared viraemia by 5 dpi, suggesting that the remaining immunocompetence in these animals was able to counteract viral replication. Furthermore, changing the route of antibody administration from intraperitoneal to intravenous might increase the biodistribution of MAR1-5A3, though the maximum volume that can be injected via this route is much lower. Additionally, this model might be further modified by co-injecting infection-enhancing flavivirus cross-reactive mouse monoclonal antibodies (e.g., 4G2) [49] to increase viral replication and disease pathogenesis.

In summary, we have shown that transient blockade of type I IFN signalling by treatment with the IFNAR1-blocking antibody MAR1-5A3 is able to promote replication of D2Y89P, but not the development of virus-induced disease, in adult wild-type mice. Our study adds to the continued efforts to develop mouse models of DENV infection and provides a tractable tool for high-throughput evaluation of next-generation vaccines and other antiviral intervention strategies.

## Figures and Tables

**Figure 1 viruses-15-00814-f001:**
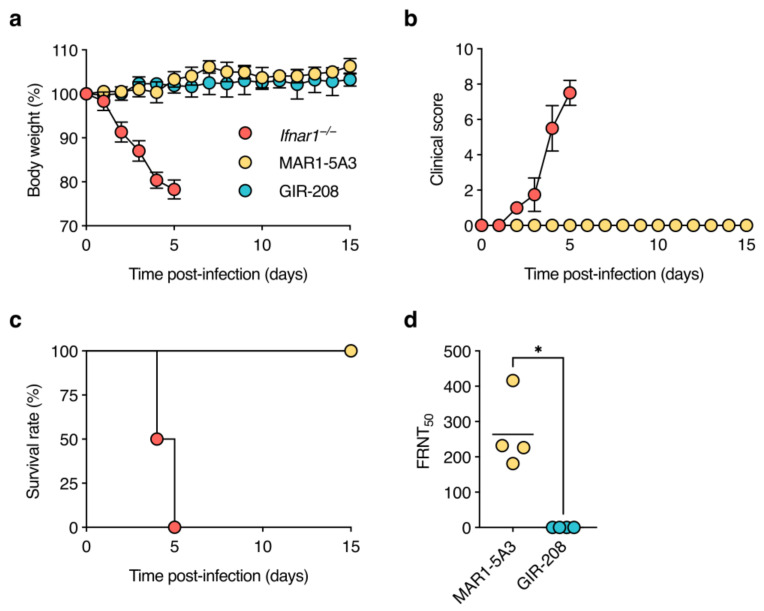
D2Y98P infection is asymptomatic in adult wild-type mice treated with IFNAR1-blocking antibody. *Ifnar1*^−/−^ mice (*n* = 4), MAR1-5A3-treated C57BL/6J mice (*n* = 4), and GIR-208-treated C57BL/6J mice (*n* = 4) were inoculated subcutaneously with 10^6^ PFU of D2Y98P. Mice were monitored daily for weight loss (**a**), clinical scores (**b**), and survival (**c**) as described in Materials and Methods. Data in (**a**,**b**) are shown as means ± standard deviation. (**d**) Sera of surviving animals were collected at 15 dpi and tested for neutralising antibodies against D2Y98P by focus reduction neutralisation test (FRNT). Reciprocal antibody titres were determined by non-linear regression analysis. Each symbol represents one mouse. Short horizontal lines indicate means. *, *p* < 0.05 (Mann–Whitney *U* test).

**Figure 2 viruses-15-00814-f002:**
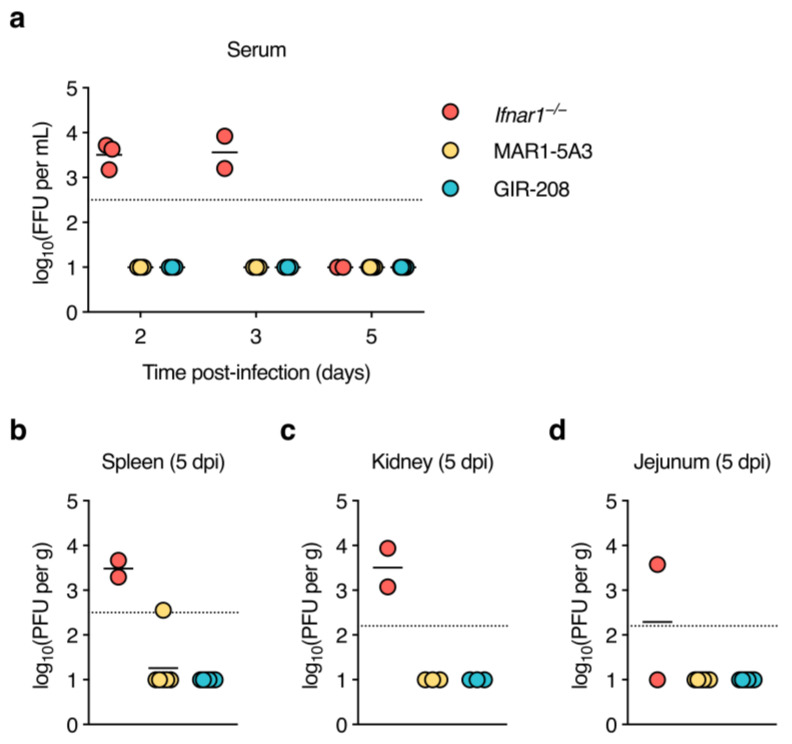
Infectious virus is undetectable in sera and organs of D2Y98P-infected adult wild-type mice treated with IFNAR1-blocking antibody. *Ifnar1*^−/−^ mice (*n* = 4), MAR1-5A3-treated C57BL/6J mice (*n* = 6), and GIR-208-treated C57BL/6J mice (*n* = 6) were inoculated subcutaneously with 10^6^ PFU of D2Y98P. (**a**) Sera were collected from alternate animals of the same group at 2 and 3 dpi and from all surviving animals at 5 dpi, and infectious virus was measured by focus-forming assay. (**b**–**d**) Organs were harvested at 5 dpi and infectious virus was measured by plaque assay in spleen (**b**), kidney (**c**), and jejunum (**d**). Each symbol represents one mouse. Short horizontal lines indicate means. Dotted horizontal lines indicate the limit of detection. Data below the limit of detection were assigned nominal values.

**Figure 3 viruses-15-00814-f003:**
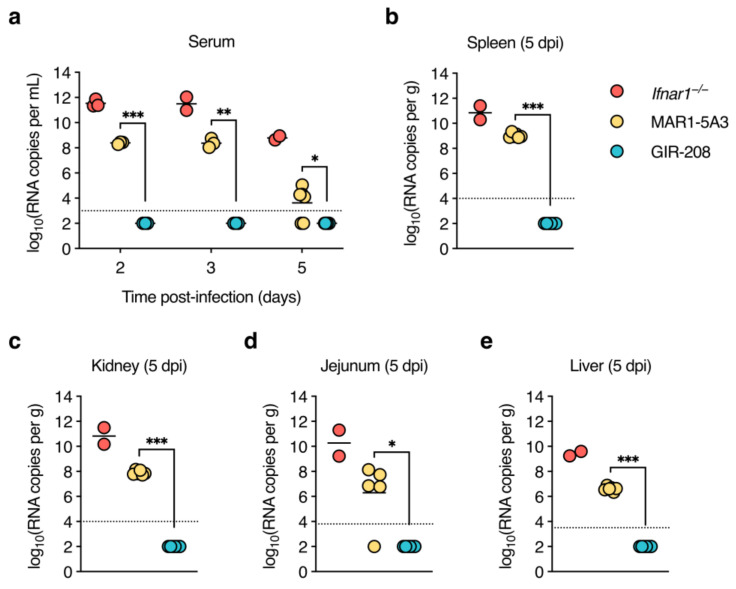
High levels of viral RNA are present in sera and organs of D2Y98P-infected adult wild-type mice treated with IFNAR1-blocking antibody. Same mouse experiment as in Figure 2. (**a**) Viral RNA in sera collected at 2, 3, and 5 dpi was quantified by RT-qPCR. (**b**–**e**) Viral RNA in organs harvested at 5 dpi was quantified by RT-qPCR. Results for spleen (**b**), kidney (**c**), jejunum (**d**), and liver (**e**) are shown. Each symbol represents one mouse. Short horizontal lines indicate means. Dotted horizontal lines indicate the limit of detection. Data below the limit of detection were assigned nominal values. *, *p* < 0.05; **, *p* < 0.01; ***, *p* < 0.001 (unpaired t-test with Welch’s correction).

## Data Availability

Not applicable.

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
