# Peer review of "Transient Blockade of Type I Interferon Signalling Promotes Replication of Dengue Virus Strain D2Y98P in Adult Wild-Type Mice"

_viruses, 2023, doi:10.3390/v15040814_

Round 1
Reviewer 1 Report
The article is quite relevant about an animal model for dengue, which is really a difficult demand for someone working in the area. Some aspects may be clearer in the manuscript.
Abstract- ok
Introduction- It would be interesting to add a summary of the results of the paper as the last paragraph of the introduction, as was done in the conclusion.
Material and methods - the authors use female animals, but most of the works use males, because of the smaller hormonal variation. Can the authors better justify why they work with females?
Can the authors explain how the 2mg antibody dosage was chosen?
Results- The authors should present some histopathological data, as this was analyzed. Add images of the liver lesions, and if any, also the spleen and kidneys.
Author Response
Reviewer 1
Abstract- ok
Introduction- It would be interesting to add a summary of the results of the paper as the last paragraph of the introduction, as was done in the conclusion.
We have added this information to the revised manuscript (lines 76–81).
Material and methods - the authors use female animals, but most of the works use males, because of the smaller hormonal variation. Can the authors better justify why they work with females?
There was no scientific rationale for using female rather than male mice. This was merely done for practical reasons as it allowed for group housing and hence reduce costs.
Can the authors explain how the 2mg antibody dosage was chosen?
2 mg has been defined as the IFNAR1-saturating dose in vivo by the group that developed this antibody, as already referred to in lines 194–195 of the Results section.
Results- The authors should present some histopathological data, as this was analyzed. Add images of the liver lesions, and if any, also the spleen and kidneys
We have not performed histopathology on any organs. We only report here macroscopic lesions that we observed of the liver and spleens during necropsy. Images of these lesions are not available.
Author Response
Transient blockade of interferon signalling promotes replication of dengue virus strain D2Y98P in adult mice Wilken et al. In this study, the authors tried to develop an immunocompetent mouse model that can be used for vaccine testing by treating wild-type mice with MAR1-5A3, an IFNAR1-blocking antibody. The manuscript is well written. They observed that antibody treatment prior to infection with a DENV2 strain results in the presence of a high level of viral RNA, but no virus particles could be detected. Infectious virus particles were detected in Ifnar1-/- mice. Also, the antibody treatment did not replicate the disease phenotypes observed in Ifnar1-/- mice. As authors pointed out, probably their method needs some modifications to develop a good immunocompetent mouse model. DENV is well known to produce defective-interfering particles. Is it possible that the viral RNA molecules detected are deleted derivatives of the wild-type RNA molecules; they did not detect any infectious particles (assayed by focus forming units), because most of them defective-interfering particles? The authors have not addressed this possibility.
We agree that defective interfering particles may likely account for the high levels of viral RNA detected and have added this to the Discussion section (lines 319–320).
Minor comments: Line 288 Replace This study… with Previous studies
We were referring to ref. 35, so a single study. We have changed the text to “That study…” and cited the study again at the end of the sentence.